# Mapping network structures and dynamics of decentralised cryptocurrencies: The evolution of Bitcoin (2009–2023)

**Marco Venturini**[1,2]*, **Daniel García-Costa**[3], **Elena Álvarez-García**[3], **Francisco Grimaldo**[3], **Flaminio Squazzoni**[1]

**1** Department of Social and Political Sciences, University of Milan, Milan, Italy, **2** Faculté des Lettres, Sorbonne Université, Paris, France, **3** Department of Computer Science, University of Valencia, Burjassot, Spain

* marco.venturini@unimi.it

**Data availability statement:** Data from the Bitcoin blockchain are available in Google

## Abstract

At the beginning of July 2025, the global cryptocurrency market capitalisation reached more than $2.8 trillion, with 1 Bitcoin exchanging for more than $105,000. As cryptocurrencies are becoming part of the global financial infrastructure, monitoring their evolution is crucial for determining whether they can be considered a sustainable long-term financial exchange system. In this paper, we have reconstructed the network structures and dynamics of Bitcoin from its launch in January 2009 to December 2023 and identified its key evolutionary phases. Our results show that network centralisation and wealth concentration increased from the very early years, following a richer-get-richer mechanism. This trend was endogenous to the system, beyond any subsequent institutional or exogenous influence. The evolution of Bitcoin is characterised by three periods, *Exploration*, *Adaptation*, and *Maturity*, with substantial coherent network patterns. Our findings suggest that Bitcoin is a highly centralised structure, with high levels of wealth inequality and internally crystallised power dynamics, which may have negative implications for its long-term sustainability.

## Introduction

Introduced by Satoshi Nakamoto in 2008 [1], Bitcoin is now worth $1.8 trillion with a daily exchange volume of $65 billion [2], excluding derivatives. In total, the entire cryptocurrency world has a market capitalisation of around $2.8 trillion. The world's largest asset management firm, BlackRock, launched a spot Exchange Traded Fund (ETF) replicating Bitcoin in January 2024, soon followed by other institutional players. By 6 November 2024, this ETF had a daily exchange volume of $97 billion [3]. The number of crypto users worldwide is estimated to be around 800 million in 2025 [4] and 1 Bitcoin is exchanged for more than $105,000 at the beginning of July 2025. While retail investors' portfolios now include substantial investments in cryptocurrencies and digital assets, Bitcoin has also recently become a key political issue, with some governments committed to Bitcoin as a strategic reserve. For example, on the day Trump won the US presidential election, the price of Bitcoin jumped from $69,500 to around

BigQuery (bigquery-public-data.crypto_bitcoin dataset). Data for replicating the outcomes of this manuscript are available at https://doi.org/10.13130/RD_UNIMI/NFURUT.

**Funding:** The author(s) received no specific funding for this work.

**Competing interests:** The authors have declared that no competing interests exist.

$76,000, showing that its value reflects political events and expectations. This suggests that Bitcoin is no longer a niche product of a group of idealistic "hippies", but has become part of the global financial infrastructure.

Beyond the bright and dark sides, the secret of Bitcoin's success is that it allows anonymous peer-to-peer transactions without any intermediary, relying solely on its decentralised consensus protocol. Understanding the complexity of this new type of economic exchange infrastructure requires a detailed study of its transaction network, made up of different components that interact in a non-trivial way. This challenge has attracted many scientists [5–9], with studies on the transaction network of Bitcoin since its early years [10,11]. Unfortunately, the research has been hampered by a limited timescale of observation. For instance, Di Francesco Maesa and colleagues [8] analysed the evolution of the Bitcoin transaction network and tested the rich-get-richer mechanism using data until December 2015. Similarly, in Alqassem et al. [7], the authors studied the structural changes of the Bitcoin network over time, but relied on data only up to September 2014. Furthermore, these studies have tended to focus on specific Bitcoin aspects to capture its network properties, such as looking at degree distributions, power laws or preferential attachment, rather than attempting a comprehensive analysis of its network formation, dynamics and evolution. Finally, while previous research has considered a longitudinal perspective to capture network dynamics, it has ignored the impact of certain important exogenous factors, such as law enforcement and public interest, with the exception of Tasca et al. [12], which linked the prevalence of business entities in the network to the Bitcoin external evolution and perception, from a prototype to a "sin" phase towards legitimate business.

Although research on Bitcoin has concentrated on multiple layers [13], here we focused on studies of the structures and dynamics of the Bitcoin transaction network that have considered transactions between addresses. We also discussed the application of this approach to the study of other blockchain-based digital assets and systems [14], in particular Ethereum [15], to show how these approaches can be extended beyond Bitcoin to other blockchain infrastructures.

In general, research attention has mostly been devoted to fitting power law distributions, using cryptocurrencies as an illustration of complex network dynamics [16]. As the literature has acknowledged the scale-free nature of the Bitcoin network, we used this point as a *fil rouge* to reconstruct the existing research on the topic. Finally, as anticipated, we focused on the phases of the Bitcoin's life, prioritising research that had already performed similar analyses.

A first study on the transaction network of Bitcoin was proposed by Reid and Harrigan [10], where they analysed blockchain data up to July 2011 and found skewed degree distributions, despite rejecting the hypothesis of a power law fit. Similarly, Ron and Shamir [17] extended the analysis by considering data up to May 2012 and found that the vast majority of addresses moved only small amounts of Bitcoin, with 98% of nodes owning less than 10 BTC. Moreover, using transactions between 2009 and 2013, Baumann and colleagues [18] found right-skewed distributions that converge to a scale-free network over time, as well as strong correlation between user activity and the USD/BTC exchange rate, and a small-world-like structure.

Alqassem et al. [7] analysed the Bitcoin transactions that took place between January 2009 and September 2014. The results showed that the network was disassortative, almost entirely connected and characterised by high wealth inequalities. Considering Bitcoin transactions up to the 23 December 2015, Di Francesco Maesa et al. [8,19] found power law degree distributions, clustering tendencies and high Gini coefficients, confirming the presence of high inequality in the network. Moreover, they found a rich-get-richer mechanism driving wealth accumulation in the network. A variation of the rich-get-richer explanation has been

proposed by Aspembitova and colleagues [20] where the authors found a good-get-richer trend, meaning that the fitness of the nodes is what drives the formation of the network. More recently, Nerurkar et al. [9] studied the Bitcoin transaction network up to 8 May 2020. Their results corroborated previous findings on clustering, disassortative and centralisation patterns. Note that these empirical patterns seem to characterise many other cryptocurrencies, not just Bitcoin. For instance, Motamed and Bahrak [21] compared the transaction networks of six cryptocurrencies over ten years, from 2009 to 2019, confirming disassortative mixing, power law degree distributions, low densities and positive clustering. Moreover, Chen et al. [22] found similar trends in three Ethereum [23] networks, and Somin and colleagues [15] found converging dynamics across eleven-thousand Ethereum-based assets and close similarities with financial markets.

Kondor et al. [11,24] studied the evolution and growth of the Bitcoin transaction network considering data from 3 January 2009 to 7 May 2013. The authors found right-skewed degree distributions that fit a power law, high Gini coefficients for both the degree and balance distributions, disassortative and clustering tendencies and a sub-linear preferential attachment mechanism. Finally, in terms of network phases, they divided the evolution of Bitcoin into two periods: an initial phase (until 2010) and a trading phase after mid-2011. With a similar empirical setting, Liang and Zeng [25] confirmed the findings of Kondor et al. [11] and extended the analysis to underline the similarities with Ethereum. In addition to the two phases identified by Kondor and colleagues [11], Tasca et al. [12] conducted a study on the dominance of different business categories along the evolution of Bitcoin. The authors studied Bitcoin transactions up to May 2015 identifying four categories: miners, gambling services, black markets and exchanges. Furthermore, considering the dominance of each category over time, they observed three distinct regimes in Bitcoin: a prototype phase until 2012, a second phase dominated by "sin" entities such as dark markets until 2013, and a third period marked by a shift from "sin" to legitimate businesses.

In summary, previous network research on the cryptocurrency markets showed a prevalence of power law distributions of degree activity and wealth, coupled with significant levels of inequality. This indicates that the Bitcoin network appears to be unequal, centralised and close to a scale-free network. Moreover, a recurring feature is the weak disassortative tendency that characterises the cryptocurrency markets.

A key limitation of previous research is the lack of recent data confirming whether previously identified patterns persist over time despite changes in investor and user characteristics and relevant external events. Most previous studies have only covered the early years of Bitcoin, rarely reaching 2015, missing nearly a decade of network expansion, shocks, and evolution (e.g. see, [11] and [8]). Furthermore, previous research has considered only a limited number of network measures. We believe that these measures must be jointly considered to improve our understanding of the Bitcoin network in the presence of complex, decentralised, and emergent patterns. The aim of this paper is to trace network patterns and describe the evolution of the Bitcoin system in more detail than previous literature has done.

Moreover, in line with Di Francesco Maesa et al. [19,26], who tested for the rich-get-richer mechanism until 2015, we verified the same mechanism with new data. This allowed us to update their analysis to assess whether the centralisation and concentration trends are due to the increasing presence of institutional investors or to the endogenous growth dynamics of Bitcoin. The addition of almost a decade of new data enables us to consider significant events that have occurred within the system and were not considered by previous studies, such as, for instance, the impact of the Covid-19 pandemic and the adoption of institutional regulations like the European Market In Crypto-Assets Regulation (MiCAR). Furthermore, while confirming previous findings on disassortative and clustering tendencies, we also investigated

the network connected components and their composition. Finally, while Kondor et al. [11] and Tasca et al. [12] have suggested to distinguish the evolution of Bitcoin into two and three periods respectively, we explored new temporal categorisations that better reflect network measures and Bitcoin history. Here, we propose a categorisation of three periods: *Exploration*, *Adaptation*, and *Maturity-* This categorisation better reflects the trends observed in our analysis and the history of Bitcoin, including important exogenous events. This categorisation has two main advantages. First, it considers novel data and overcomes the predominance of strict business typologies as in Tasca et al. [12]. Second, it considers both internal network trends and external dynamics, highlighting their interdependence and enriching our understanding of the endogenous and exogenous drivers of the system's evolution.

## Methods

We retrieved all Bitcoin blockchain data from the 1 January 2009 to the 31 December 2023, where each record collected is a block of transactions with input and output addresses, amounts sent and received, a block number and a timestamp. We also included a dummy indicating the existence of a special transaction -with no input address- called Coinbase, which represents the reward for mining the block. This allowed us to collect all time-ordered exchange events, sending or receiving, that have occurred since the inception of Bitcoin across the entire population of addresses. We used this data to perform a longitudinal network analysis of the complete on-chain transactions.

Following Kondor et al. [11] and Alqassem et al. [7], we reconstructed the address-to-address edges (the *Address network*, as defined by Wu and colleagues [27]) and the proportional weights based on the amount of tokens moved by each address, as follows:

$$v^{(i \to j,n)} = \left( v_{\text{in}}^{(i,n)} - \frac{t_{\text{fee}}^{(n)} \cdot v_{\text{in}}^{(i,n)}}{t_{\text{in}}^{(n)}} \right) \cdot \frac{v_{\text{out}}^{(j,n)}}{t_{\text{out}}^{(n)}} \tag{1}$$

where $v^{(i \to j,n)}$, the value, is the weight of the edge -the number of tokens transacted- between nodes $i$ and $j$ in transaction $n$; $v_{\text{in}}^{(i,n)}$ is the amount sent by node $i$ within transaction $n$; $t_{\text{in}}^{(n)}$ is the total volume sent in a transaction $n$ by all the participating addresses, and $t_{\text{fee}}^{(n)}$ is the total fee of transaction $n$. We used the same notation $v_{\text{out}}^{(j,n)}$ and $t_{\text{out}}^{(n)}$ referring to the output, where $j$ refers to the receiving node.

To obtain a more manageable dataset, we encrypted input/output addresses and transaction hashes, converting public key addresses to integer IDs to reduce storage requirements and improve usability. We maintained a dictionary mapping the new IDs to the original public keys for traceability. This allowed us to obtain a dataset of approximately one billion unique transactions and 11.5 billion directed address-to-address transactions, including all the network activity since the first ever transaction, the Genesis transaction, which was initiated by Satoshi Nakamoto on 3 January 2009. We obtained an edge list representing directed transactions, with associated timestamps and token amounts (see Fig 1 for our data pipeline).

We conceived the Bitcoin system as a directed network $G = \{V, E\}$ where $V$ and $E$ are the sets of nodes and edges, respectively. Following Nerurkar and colleagues [9], we considered "value" and "timestamp" as edge attributes. Since multiple transactions can occur between the same addresses, $G$ is a directed multi-graph. Furthermore, since $G$ is a multi-graph, we were able to group all the transactions going in the same direction under the same pair of nodes. It follows that "value", the attributes of the edges indicating the amount of tokens transferred, becomes the arithmetic sum of all the edges within a pair of nodes per year. To account for the otherwise missing in- and out-movements, we developed a second edge attribute called

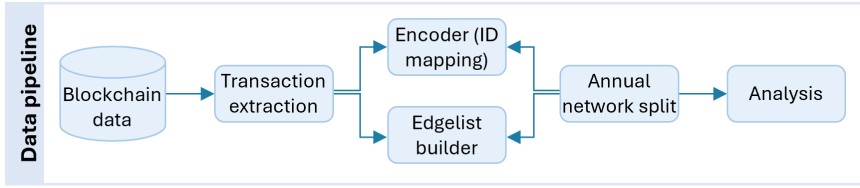

**Fig 1. The data pipeline.**

"activity", which counts the number of times each transaction between the same pair of nodes is repeated in a year. Our graph object is formulated as follows:

$$G_t = \{V_t, E_t, w_1 : E_t \to \mathbb{R}_{>0}, w_2 : E_t \to \mathbb{N}^*\} \tag{2}$$

where $G_t$ is a weighted graph with only positive weights $w_1$ and $w_2$, corresponding to "value" and "activity", respectively. In our empirical setting, we choose fifteen consecutive snapshots of the network, $t = \{1, \ldots, n\}$, $n = 15$, each corresponding to a calendar year, thus capturing the dynamics and evolving patterns that would otherwise be flattened into a static representation.

Given the variation in transaction size and the presence of noise, we set a threshold below which transactions were excluded from the analysis. We set it at 0.0001 bitcoins per year, meaning that any edge that moves less than (or equal to) 0.0001 bitcoins between two addresses in a full year is removed from the network. This allowed us to solve two problems: first, we removed so-called "dust" transactions [28], which are either unintentional or deliberate attacks on an address; second, we did not consider annual movements that have no economic relevance. We performed robustness checks to confirm our choice and the results were consistent (for details, see the Results section). The network can be finally defined as follows:

$$H_t = \{V'_t, E'_t, w_1 : E'_t \to \{w_1 \in \mathbb{R} \mid w_1 > 0.0001\}, w_2 : E'_t \to \mathbb{N}^*\} \tag{3}$$

where $H_t$ is a subset of the weighted graph $G_t$, with weight $w_1$ only greater than 0.0001 bitcoins. Respectively, $V'_t$ and $E'_t$ are the subset of nodes and edges at time $t$. We have also removed Coinbase transactions and self-loops generated by the address change operation from the network for the sake of consistency. These are system artifacts that have no substantive meaning when the focus is on the flow of tokens between different addresses.

As we wanted to provide a comprehensive overview of the system evolution, we measured both global and local network metrics. These metrics include degree distributions, connected components, assortative mixing tendencies and Gini indices to assess inequality across distributions. As centralisation and concentration trends are the main focus of our research, we also examined the rich-get-richer pattern [8]. Following Di Francesco Maesa et al. [8,19], we tested two hypotheses. First, the richest nodes increase the share of wealth and in-degree activities they control. Second, those who control most of the wealth and in-degree activities tend to remain the same over time. We adapted Eqs (4) and (5) to our empirical case as follows: since $b(v)$ represents the cumulative balance of a node $v$ over time $t = \{1, \ldots, n\}$, $n = 15$, we calculated the annual balance by taking the total amount received in Bitcoin, then subtracting the total amount sent, adding any Coinbase transactions ($\beta_t(v)$), and subtracting the annual fees paid ($\alpha_t(v)$). The last two terms -Coinbase transactions and annual fees- are implicitly included in our initial edge formulation. Finally, we also added the cumulative

wealth from the previous years to the calculation ($\theta_{t-1}(v)$) as follows:

$$b_t(v) = \sum_{(u,v) \in E'_t} w_1(u,v) - \sum_{(v,u) \in E'_t} w_1(v,u) - \alpha_t(v) + \beta_t(v) + \theta_{t-1}(v) \tag{4}$$

Similarly, the in-degree richness $i(v)$ of a node $v$ accounts for the cumulative in–degree at time $t$, where the first term is the in-degree at $t$ and $\gamma_{t-1}(v)$ is the accumulated in–degree of node $v$ until $t-1$ as follows:

$$i_t(v) = \sum_{(u,v) \in E'_t} w_2(u,v) + \gamma_{t-1}(v) \tag{5}$$

Here, unlike the previous part of the analysis, we have included both Coinbase transactions and self-loops, as the focus shifted from the exchange between addresses to wealth accumulation. This was the same test as Di Francesco Maesa and colleagues [8], but with data after 2015. As mentioned above, the system has evolved significantly since then, with the entry of new institutional actors and private investors who have challenged the status quo and accumulated knowledge. The first hypothesis is tested by considering the ten ($k$) richest nodes, in terms of balance and in-degree, in the set $V$ at time $t$. We used two ratios consisting of the wealth (or in-degree) of the ten richest addresses over the total wealth (or in-degree) of the network. The higher the $r(b)$ and $r(i)$, the higher the inequality with respect to the full set of nodes (see Eq (6)).

$$r_t(b) = \frac{\sum_{v \in B_{k,t}} b_t(v)/k}{\sum_{v \in V_t} b_t(v)/|V_t|}, r_t(i) = \frac{\sum_{v \in I_{k,t}} i_t(v)/k}{\sum_{v \in V_t} i_t(v)/|V_t|} \tag{6}$$

The second hypothesis is evaluated by measuring the variability within the set of the ten ($k$) richest nodes, specifically, using union sets. We have calculated the cumulative number of addresses in the set of the richest node at time $t$ over all the years of observation: the theoretical maximum is given by $k * t$ which amounts to $10 * 15 = 150$. We have calculated the quantities of interest, $X_{b,t}$ and $Y_{i,t}$ in Eq (7), as in [8], where $B_k^j$ and $D_k^j$ are, respectively, the set of the $k$ richest nodes, at time $t$, in terms of balance and in-degree activity. If both quantities are consistently found below the theoretical maximum, then, our hypothesis is confirmed. Note that we have used the same notation as Di Francesco Maesa et al. [8] to improve comparability.

$$X_{b,t} = \left| \bigcup_{j=1}^{t} B_k^j \right|, Y_{i,t} = \left| \bigcup_{j=1}^{t} D_k^j \right| \tag{7}$$

## Results

This section presents our results with particular emphasis on global and local measures, and the test of the "rich-get-richer" pattern.

### Global and local measures

We started by characterising the network growth over time. Tables 1 and 2 show the number of nodes and edges for each of the fifteen snapshots considered. The growth of the network is rapid, starting with 2873 nodes and 3500 edges in 2009 and reaching 148 million nodes and 568 million edges in 2023. The scale of this expansion is remarkable and highlights the success that Bitcoin has had over the years. Given the size of the network, its low density is not

**Table 1. Summary statistics of the network snapshots: number of addresses, edges and edge density from 2009 to 2016.**

| Year | 2009 | 2010 | 2011 | 2012 | 2013 | 2014 | 2015 | 2016 |
|---|---|---|---|---|---|---|---|---|
| # Address | 2873 | 122183 | 2320838 | 5948985 | 15990181 | 34185739 | 55780129 | 94962579 |
| # Edges | 3500 | 176946 | 5477578 | 16944653 | 52174726 | 240273864 | 297597762 | 314446217 |
| Density | 4.24e-05 | 1.19e-08 | 1.02e-08 | 4.79e-07 | 2.04e-07 | 2.06e-06 | 9.56e-06 | 3.49e-06 |

**Table 2. Summary statistics of the network snapshots: number of addresses, edges and edge density from 2017 to 2023.**

| Year | 2017 | 2018 | 2019 | 2020 | 2021 | 2022 | 2023 |
|---|---|---|---|---|---|---|---|
| # Address | 143253859 | 117644156 | 130922321 | 166858661 | 167060474 | 151005961 | 148245334 |
| # Edges | 560461106 | 430264236 | 565379186 | 658268828 | 597585938 | 542948094 | 567921141 |
| Density | 2.73e-06 | 3.11e-06 | 3.30e-06 | 2.36e-06 | 2.14e-06 | 2.38e-06 | 2.58e-06 |

very surprising: a network with a billion of unique nodes is rarely dense. Interestingly, Tables 1 and 2 show that edge density stabilised after 2015, indicating that the system reached a more stable number of transactions per node. This is confirmed by the mean degree (Fig 4a), which stabilises after the same year, probably due to the increased maturity of Bitcoin.

In Methods section, we filtered the data to avoid noise in the network and to obtain more consistent results. We ran some tests to confirm that filtering out the transactions with a cumulative annual value of less than 0.0001 bitcoins did not bias our analysis. Table 3 shows the share of the Bitcoin volume considered with the filter to that of the whole network. The values range between 0.99 and 1, meaning that we have (approximately) considered the total Bitcoin volume even when applying the filter. Similarly, the share of nodes considered indicates that we were able to account for the vast majority of nodes in the network after removing noise and non-economically significant transactions. Note that the "NA" values are due to the remarkable size of the dataset for these specific years and computational issues, but we are confident that the numbers are consistent with other years.

We also computed the in-degree and out-degree of nodes weighted by activity and value, as specified in Methods section, along with the respective distributions. To test for network centralisation and concentration, we examined the degree distributions. Figs 2 and 3 show two different years, one representing the early years of activity and the other the last period of observation. The four distributions for 2011 and 2023 show a clear pattern that has emerged since the early stages of the system's development: transaction volume is concentrated in the hands of a small number of addresses, while the majority of nodes only manage a few exchanges per year. This trend remains stable throughout the period up to 2023 (Fig 3), despite the growing size of the network.

We also looked at the higher moments of the distributions: the average activity-weighted degree has stabilised since 2015, while the average value-weighted degree has decreased as the price of Bitcoin has risen (see Fig 4a), and the standard deviations followed the trends. The decrease occurred because, while Bitcoin is still made up of the same units -satoshi- the

**Table 3. Share of the Bitcoin volume and nodes considered when filtering out transactions with an annual cumulative value of less than 0.0001 BTC.**

| Year | 2009 | 2010 | 2011 | 2012 | 2013 | 2014 | 2015 | 2016 | 2017 | 2018 | 2019 | 2020 | 2021 | 2022 | 2023 |
|---|---|---|---|---|---|---|---|---|---|---|---|---|---|---|---|
| BTC Considered | 1 | 1 | 1 | 1 | 0.99 | 0.99 | 0.99 | 0.99 | 0.99 | 0.99 | 0.99 | 0.99 | 0.99 | 0.99 | NA |
| Nodes Considered | 1 | 0.99 | 0.89 | 0.99 | 0.97 | 0.98 | 0.96 | 0.98 | 0.96 | 0.92 | 0.95 | 0.96 | NA | 0.94 | NA |

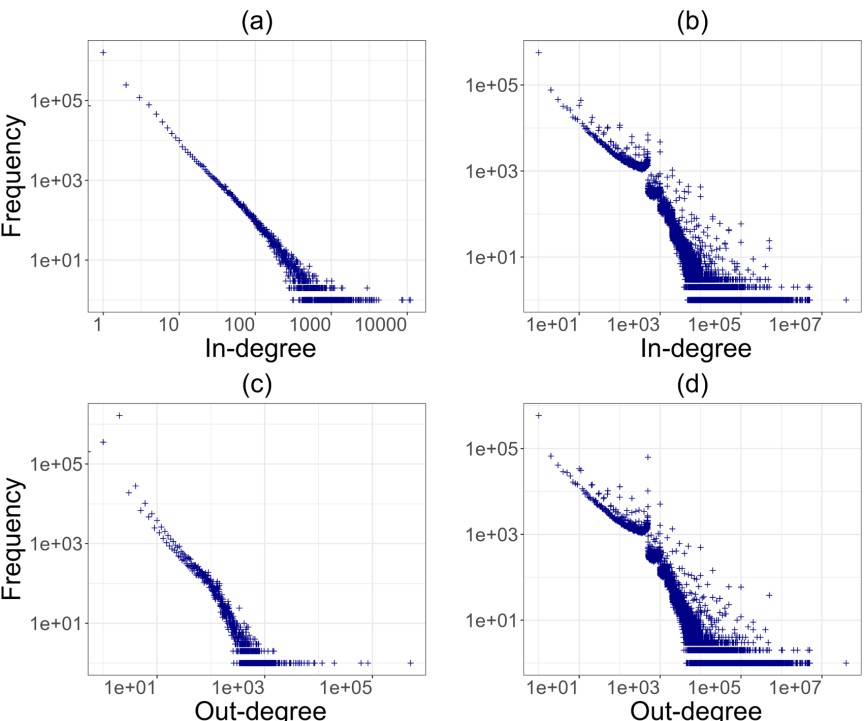

**Fig 2. The four degree distributions in 2011: (a) in-degree weighted by activity, (b) in-degree weighted by value, (c) out-degree weighted by activity, and (d) out-degree weighted by value.** The observed degree refers to individual Bitcoin addresses, while the weights -value and activity- represent, respectively, the total amount transacted and the total number of transactions between two addresses over the course of a full year.

unit prices have increased over time, reducing the number of tokens that need to be moved to transfer the same value in dollar terms. Skewness and kurtosis have increased over time, indicating increasing centralisation, unevenness and heavier tails, to stabilise again after 2015. It is important to note that the observed centralisation does not indicate a unique evolutionary process, but rather mimics trends common in other complex network infrastructures [16,29], such as technological networks.

To measure the inequality in the network, we computed Gini coefficients for the four degree distributions: all were above 0.75 after 2011-2012, with those associated with value-weighted distributions approaching 1. Fig 4b shows higher Gini values for the two distributions related to value, i.e., the value of transactions sent or received. We might have expected this pattern because, while completing a transaction is easy and relatively inexpensive, moving a significant amount of tokens requires great resources. This can be explained by the large volume of transactions that large addresses complete each year; they require resources and facilities that smaller participants do not have, which explains the greater inequality between the two distributions. Finally, due to the small number of addresses and the popularity of Coinbase transactions, which accounted for the majority of transactions at the time, the out-degree distributions in 2009 have lower Gini coefficients than their in-degree counterparts.

Fig 5a shows the results of an analysis of assortative mixing [30,31] and clustering motifs to track homophily and clustering within the network. Specifically, we computed degree assortativity and the transitivity index. We found that the degree assortativity coefficient was low but negative over the years, and stabilised again after 2015. This indicates a general tendency for high degree nodes to be connected to nodes that transact with fewer other nodes.

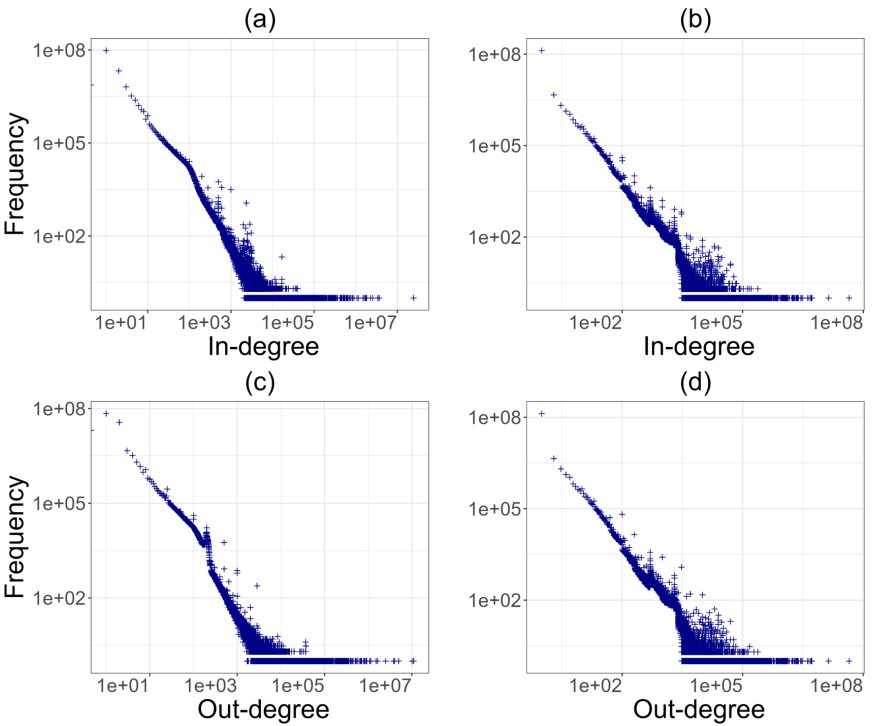

**Fig 3. The four degree distributions in 2023: (a) in-degree weighted by activity, (b) in-degree weighted by value, (c) out-degree weighted by activity, and (d) out-degree weighted by value.** The observed degree refers to individual Bitcoin addresses, while the weights -value and activity- represent, respectively, the total amount transacted and the total number of transactions between two addresses over the course of a full year.

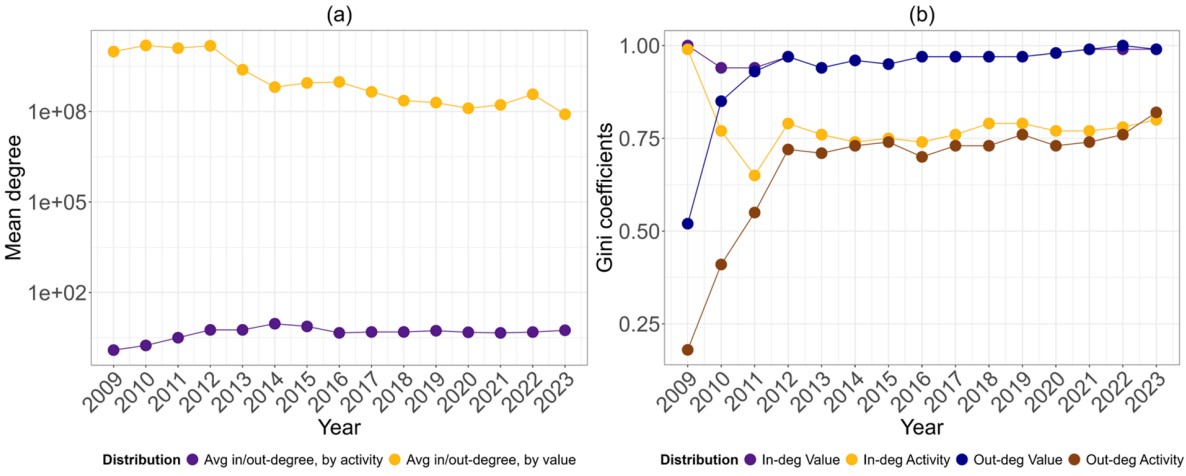

**Fig 4. (a) Average degree weighted by activity and value.** In this case, in- and out-degree averages are equal. (b) Gini coefficients of the four degree distributions.

Fig 5b shows the transitivity coefficient, i.e. the tendency of the nodes to form clusters, which we measured using the average local clustering coefficient to avoid biased measures due to network sparseness. The coefficient was found to be positive, albeit small, and to stabilise after 2014, consistent with previous findings [11] in both magnitude and direction.

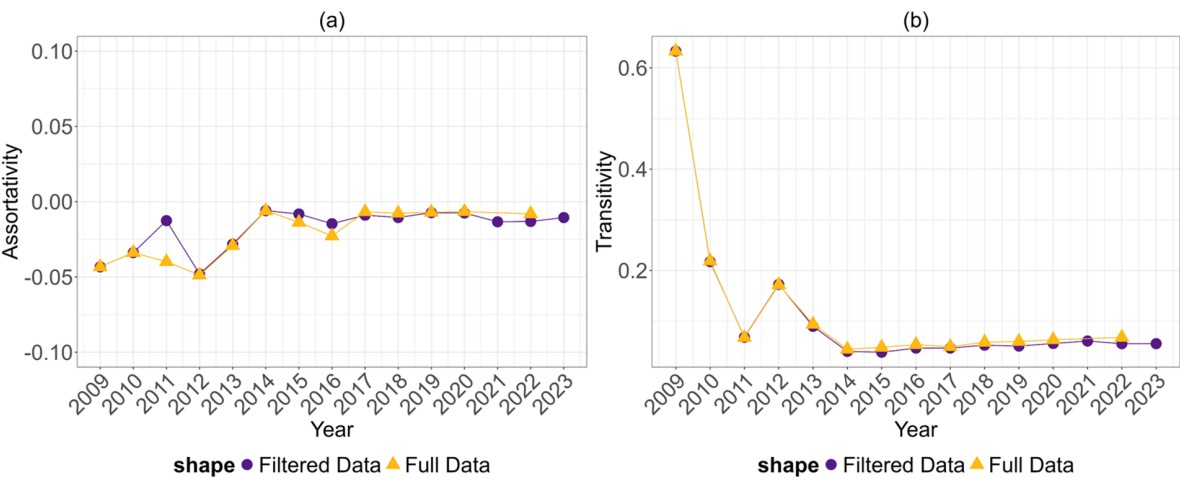

**Fig 5. (a) Effect of the filter on Degree Assortativity and (b) Transitivity Coefficients, per year.**

Such transitivity suggests two possibilities: either addresses tend to form triadic structures, or the network operates as a closed system in which tokens circulate among the same groups of nodes.

We also plotted the assortativity and transitivity indices for the full data (as in Methods section). Fig 5 shows that the filtered and full measures are consistent across the two different datasets. If anything, the small variation we observe assortativity in 2011 strengthens our strategy. This means that we have not overestimated any outcome; at most, we have underestimated some effects in the early years of the network. This confirms the effectiveness of our filter.

We also examined the connected components of the network *G*, distinguishing between "weakly" and "strongly" connected components. In a weakly connected component, edge directions are disregarded, whereas in a strongly connected component, they are considered [32]. Specifically, within a weakly connected component, each pair of nodes is connected by some path when edge directions are ignored. In contrast, in a strongly connected component, there is a directed path from any node to any other node within the component following the edge directions. We found that the largest weakly connected component (LWCC) includes almost all addresses since 2011, indicating that approximately every node can be reached by every other node by either receiving or sending tokens. Thus, despite the sparsity of the network, every node is indirectly connected to the entire set of nodes.

On the other hand, the Largest Strongly Connected Component (LSCC) does not include all addresses but a significant proportion of nodes. The proportion of nodes in the LSCC peaked in 2014, the year of the Mt. Gox crisis, and has since fluctuated inversely with the annual returns for 2015-2017 and 2021-2023, while correspondingly with the returns for 2018-2020. To corroborate our findings on the LSCC and LWCC, we computed Gini indices on the size distributions of the two types of connected components. Fig 6a shows that the Gini index for the WCCs has been consistently close to 1 since the network's inception, indicating a high degree of inequality in the size distribution. Similarly, the Gini index for the SCCs remains above 0.75, albeit with fluctuations that mirror the behaviour of the LSCC. It is worth noting that the network lacks any relevant completely separate component or multiple centres of activity. Indeed, everything revolves around a giant component that is entirely reachable through indirect connections: a feature that was neither explicitly designed nor anticipated. In

addition, there is a substantial strongly connected component that tends to shrink as Bitcoin's value increases.

Fig 6b shows the 1% of the nodes with the highest in-degree and out-degree, including the share of their controlled in-edges and out-edges. Despite fluctuations, these shares converge around 0.5, indicating that the most active 1% control about half of the network's transactions. This concentration peaked between 2011 and 2012, declined until 2014, and increased from 2015 onwards, reducing the range of oscillations. These years were significant: the former marked the end of the exploratory phase, while the latter represents a pivotal moment following the shutdown of SilkRoad by the FBI in October 2013 and the crisis of Mt. Gox, the oldest and once largest cryptocurrency exchange, in 2014, leading to a more mature phase starting in 2015.

We also measured the annual percentage of these 1% of nodes present in the LSCC and LWCC over time (see Fig 6c). For the LSCC, these percentages were initially low, but increased significantly, peaking in 2014 and stabilising thereafter. The percentage of the richest nodes by out-degree in the LSCC was higher than that by in-degree, indicating that the

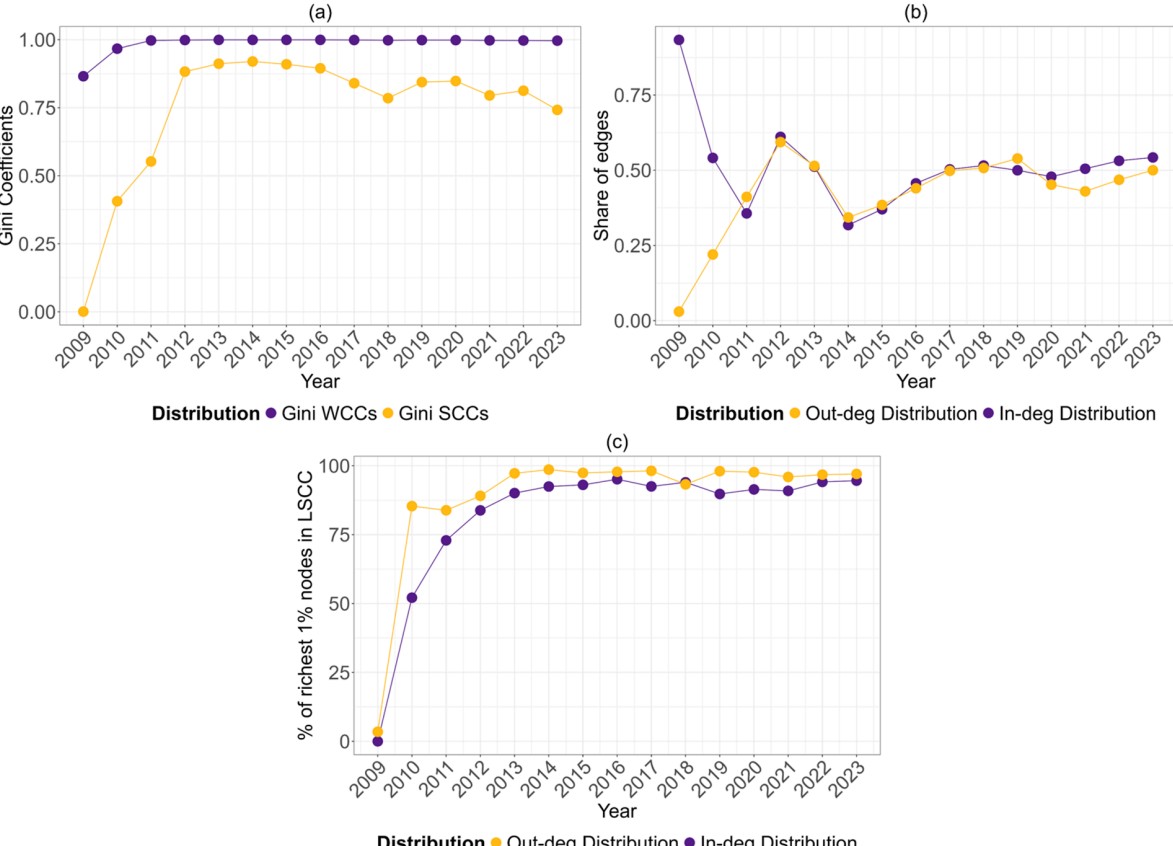

**Fig 6. (a) The Gini index of the size distribution of the Weakly and Strongly Connected Components.** This index measures the inequality in the size distribution of the connected components within the network, and shows that most of the addresses are part of a single giant component that dominates the system. (b) Shares of the in- and out-egdes controlled by the richest 1% of nodes in terms of in- and out-degree. This measures the proportion of all incoming and outgoing edges that are controlled by the top 1% of addresses based on their in-degree and out-degree. (c) The percentage of the richest 1% in terms of in- and out-degree that are present in the Largest Strongly Connected Component. This shows that the set of the most connected nodes is within the core of the network.

LSCC includes more of the most active out-degree nodes (senders) rather than in-degree nodes (receivers). This is expected, as high out-degree nodes are crucial for network connectivity, while high in-degree nodes may receive most of their information from a few other nodes and remain outside the strongly connected component.

Finally, we identified three main periods in the life of the Bitcoin network: from 2009 to 2012 the system was in the *Exploration* phase, from 2012 to 2015 in the *Adaptation* phase and from 2015 in the phase of *Stabilisation* phase. This periodisation is based on two different aspects, namely the trends of the measures we have presented and the timing of the main Bitcoin events. On the one hand, most of the measures we observed fluctuated strongly in the Exploration phase, showed some rapid changes in the Adaptation phases and tended to stabilise from 2015 onwards. On the other hand, from the beginning until 2012, the system was in its early phase with few participants and Bitcoin only reached a dollar value in 2011; 2012 marked the end of this phase with the first halving of Bitcoin. Between 2012 and 2015, several events characterised Bitcoin: SilkRoad was shut down by the FBI in 2013, Mt. Gox went bankrupt in February 2014, and in the same year the US Internal Revenue Service started to consider Bitcoin-related gains and losses as reportable assets; in this phase, Bitcoin adapted to greater participation and external influence. Finally, we identified the maturity phase as beginning in 2015, when volatility, while still significant, gradually decreased, institutional participation increased, and Bitcoin began to shed its stigma as a vehicle for illicit activity.

## Rich-get-richer

For the rich-get-richer mechanism,we first examined the increase in wealth controlled by the richest ten ($k$) nodes over time, and then the change in the composition of the set of richest nodes. We found substantial evidence to support both hypotheses: the richest nodes increased their share of wealth and in-degree activity, while the addresses controlling the most money and in-degree activity remained relatively stable.

To calculate $r_t(b)$ and $r_t(i)$ (see Eq (6)), the wealth (or in-degree) controlled by the $k$ nodes relative to the whole network, and to measure $X_{b,t}$ and $Y_{i,t}$ (see Eq (7)), indicating the variation in the richest set, we de-anonymised most of these addresses using data from BitInfoCharts, Arkham Intelligence, and the datasets from [33,34]. These sources typically compile publicly available information about major cryptocurrency addresses through web scraping, blockchain analysis, and entity tagging. Although de-anonymisation remains inherently imperfect due to the pseudonymous nature of blockchain technology, combining multiple independent data sources improves the reliability of entity attribution. Please note that we were unable to de-anonymise some of the nodes from the early years due to insufficient information. However, this actually strengthens our results, as we have already accounted for all possible address heterogeneity in the early years.

The curves shown in Fig 7 reveal a clear pattern of growth in the ratios $r_t(b)$ and $r_t(i)$ over time, thus, indicating an increasing concentration of wealth among the ten richest nodes, both in terms of wealth and in-degree. Specifically, Fig 7 shows that the set of $k$ nodes is richer at time $t$ than at *t-1* with respect to the whole network.

Fig 8a and 8b show, respectively, the evolution of the union set of the richest nodes over the 15 snapshots and its theoretical maximum. The curves of the observed values remained clearly below the expected maximum lines. In particular, the stability in the control of in-degree activity exceeds that of wealth accumulation; this could be explained by the fact that it is easier to accommodate more incoming transactions than to significantly increase the total wealth accumulated, so that a node can receive numerous transactions of low value and be among the top $k$ nodes only for the in-degree statistics.

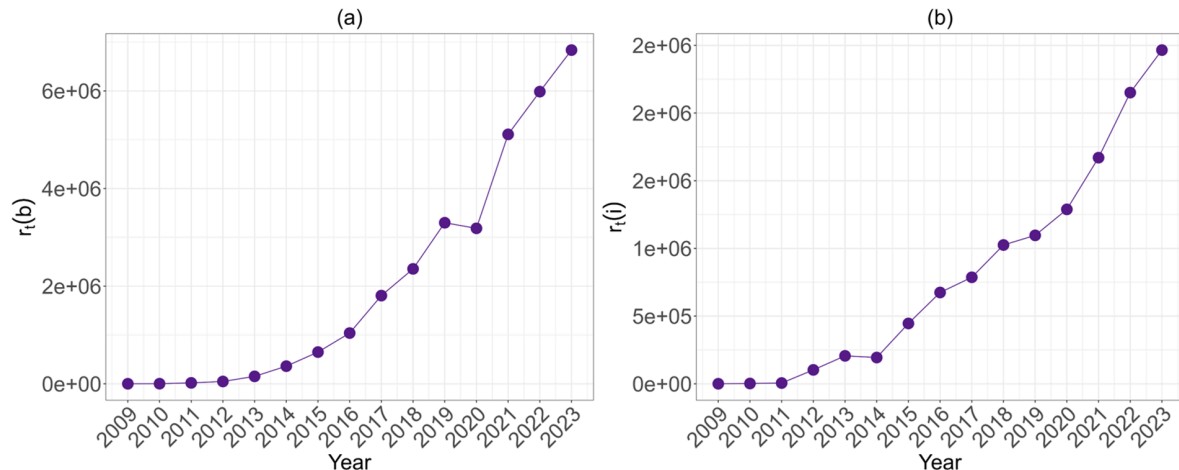

**Fig 7. (a) The evolution of $r_t(b)$ and (b) $r_t(i)$ over time; $r_t(b)$ and $r_t(i)$ are defined as in Eq (6).** $r_t(b)$ represents the ratio between the average balance (in bitcoins) of the ten richest nodes and the average balance across the entire network, while $r_t(i)$ refers to the same ratio but based on in-degree. An increase in either metric indicates that the wealthiest nodes -whether in terms of balance or connectivity- are becoming richer or more connected relative to the rest of the network.

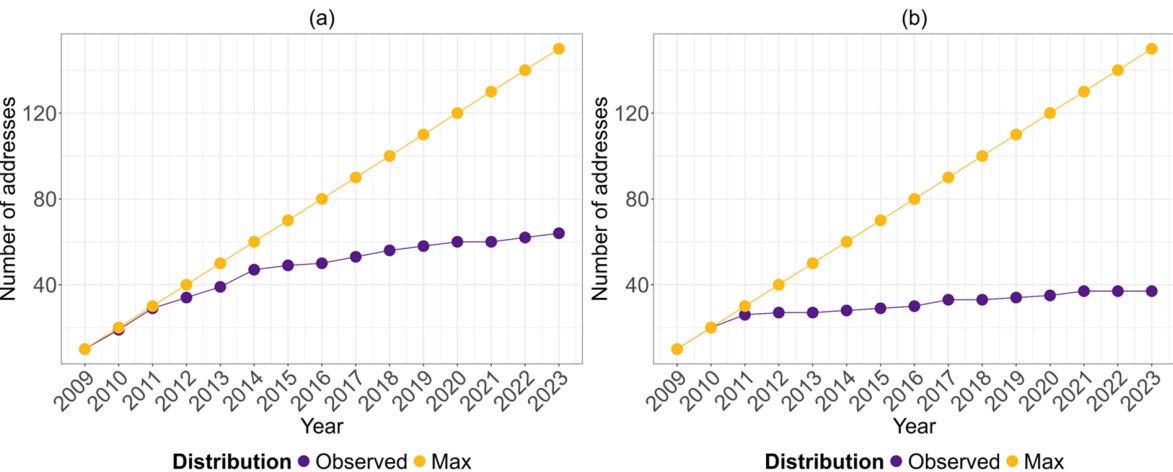

**Fig 8. (a) The evolution of $X_{b,t}$ and (b) $Y_{i,t}$ and maximum possible values over time; $X_{b,t}$ and $Y_{i,t}$ are defined as in Eq (7).** A curve that falls below the theoretical maximum indicates a persistence of the richest nodes over time, meaning that the same entities tend to remain among the richest addresses in different periods.

This indicates that the richest nodes tend to maintain their status throughout the period, they do not lose their top ten placement over time.

Fig 9a and 9b show, respectively, the evolution of the sets of the richest addresses in terms of balance and in-degree. In particular, both figures show the persistence of the same addresses among the richest, e.g. a wallet linked to Mt. Gox (see Fig 9a) has been in the top 10 since 2011. Moreover, the nodes that emerge as the richest are increasingly exchanges that act as intermediaries in the system. Finally, as noted for Fig 8, we observe less variation in terms of in-degree richness than in terms of balance as it is much easier to receive incoming transactions than to accumulate wealth.

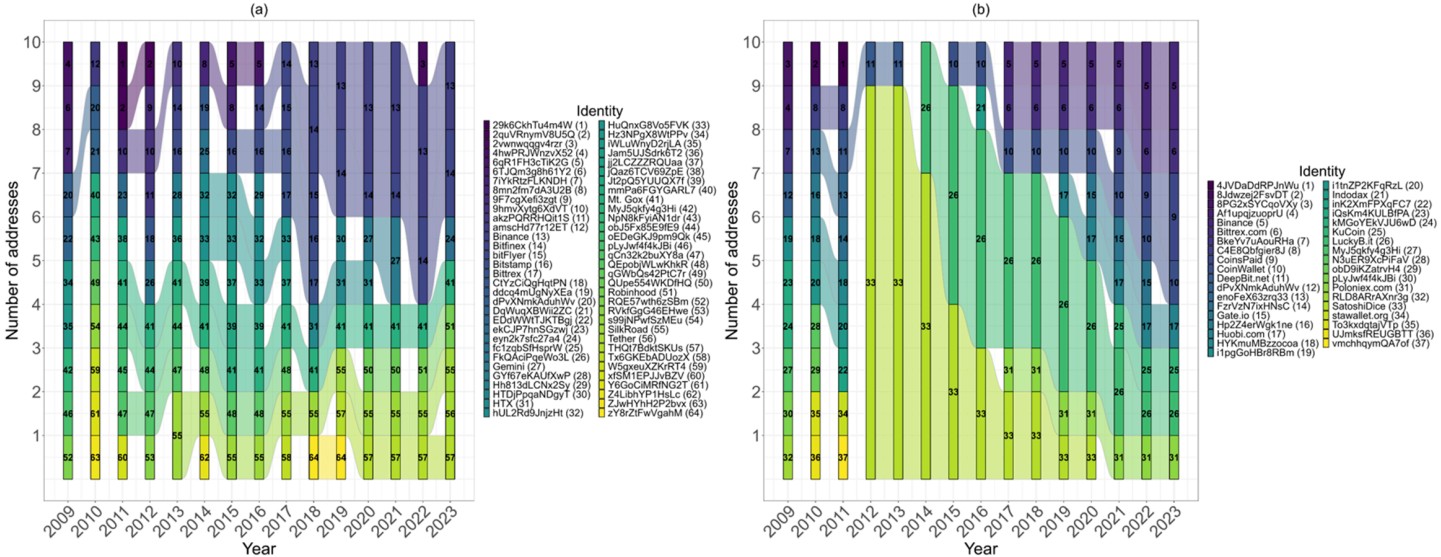

**Fig 9. Evolution of the set of the richest nodes over time, (a) by balance and (b) in-degree.**

Overall, these results provide clear support for the rich-get-richer mechanism found by Di Francesco Maesa et al. [8] up to 2015. Furthermore, we argue that this mechanism contributed to the creation of path dependencies in the system: once a node enters the richest group, it tends to stay in it and continue to accumulate more resources. It is important to highlight that these dependencies persist even after the entry of institutional actors, as the composition of the union sets had already started to stabilise by 2014, before any significant institutional involvement.

## Discussion

The creation of Bitcoin marked a turning point in the evolution of the financial systems and institutions. Characterised by decentralisation, trustless transactions, and an alleged egalitarianism [35], the evolution of Bitcoin has revealed a puzzling path: centralisation tendencies have become apparent [11,17], systemic bottlenecks have emerged and Bitcoin has started to be populated by institutional actors and retailers, as well as being subject to increasing regulation.

In our study, we wanted to explore the unplanned evolution of Bitcoin's network structures and dynamics, focusing on centralisation trends, network characteristics, and testing the rich-get-richer mechanism. We first characterised the Bitcoin transaction network by mapping its structures and dynamics over fifteen one-year snapshots. The longitudinal approach then allowed us to study evolutionary trends from the system's inception to its most recent developments. We performed a network analysis that revealed low density values -typical of large-scale networks- similar to what Nerurkar and colleagues [9] found in their study.

Furthermore, in line with previous research, we found right-skewed degree distributions indicating a pronounced centralisation of node activity and wealth accumulation as in Kondor et al. [11,24] and Di Francesco Maesa and colleagues [7,8], with a minority of players dominating the network. In particular, we focused on centralisation and concentration trends, which are contrary to expectations for a system like Bitcoin. As in previous studies [7, 10,11,18], we found skewed distributions and high levels of inequality from the very first years

of the system's life, since what we defined as the *Exploration* phase, a period characterised by early adopters and low levels of activity. Therefore, we excluded any role of external regulation and institutional actors in the emergence of the centralisation trend: it emerged endogenously from the free interactions between addresses in the network. Our results showed that these distributions were highly unequal, with Gini indices ranging from 0.75 to 1, in line with earlier studies [7,8,11].

We also found evidence for the rich-get-richer mechanism, which we investigated through the accumulation of nodes' wealth and activity and its persistence over time. While this was already found for the first years of Bitcoin activity by Di Francesco Maesa et al. [8], our results confirm that this pattern has even strengthened over time. This path-dependent behaviour is indicative of a system in which initial advantages are reinforced over time, shaping the emergent properties of the network. Once an early node becomes one of the richest, it is rarely displaced. Therefore, our analysis revealed the early onset of path dependencies behind these network trends. The immutable nature of the blockchain may facilitate this path dependency, while the limited and diminishing supply of new tokens may have encouraged wealth accumulation among early network participants. However, this should not be taken for granted; the growing role of financial institutions, defined by Baldwin [36] as the corporatisation of Bitcoin, should not be neglected. This is confirmed by the small but significant changes in the set of the richest nodes (see Rich-get-richer section): over time, the richest nodes are increasingly institutions rather than anonymous owners or early adopters.

We found persistent disassortative mixing tendencies, thus confirming previous evidence [7,11,21,27]: although small, these tendencies are indicative of a system in which low-connected nodes preferentially connect with high-connected nodes. This pattern suggests a preference for trading with one's opposite in terms of network activity. Furthermore, while a positive transitivity coefficient is not new in Bitcoin [11], we found a slightly stronger clustering tendency compared to previous results [9]. The clustering, although low, warrants careful consideration in future studies, as it may imply a variety of triadic configurations relevant for understanding and explaining the network evolution.

Focusing on the network components, we found that almost the entire network is indirectly connected, with no relevant disconnected components. This confirms the findings of Alqassem et al. [7] and Nerurkar and colleagues [9], who also found a giant component where the majority of nodes were at least indirectly connected. We also found a large strongly connected component that emerged as a focal point in the network; extending [7,9], we observed a higher proportion of out-degree rich addresses than in-degree rich nodes in it, as the former are more important for keeping this network component fully connected.

Finally, by studying the dynamics of our measures and the history of Bitcoin, we defined three periods in the evolution of Bitcoin: the *Exploration* phase, the *Adaptation* phase, and the *Maturity* phase. The exploration phase was characterised by early adopters and low activity levels until the first halving (2012), a period in which Bitcoin incubated its disruptive evolution; the adaptation phase saw the shutdown of Silkroad -one of the most famous dark markets [37]-, the collapse of Mt. Gox (2014) -the largest exchange at the time- and the first regulatory attempts by the US Internal Revenue Service. The maturity phase then began in 2015 and was characterised by decreasing volatility, increasing activity and participation, and greater institutional involvement. During this phase, not only was Bitcoin less stigmatised than before, but we also observed a stabilisation in our network measures after 2015, meaning that the Bitcoin internal functioning and its external evolution are linked. In addition to Kondor and colleagues [11] and Tasca et al. [12], considering these two aspects together -metric dynamics and Bitcoin-related events- is key to capture the complexity of network systems such as Bitcoin and cryptocurrencies in general.

However, Bitcoin and its blockchain are too broad to be captured in a single study. For example, the timestamped nature of Bitcoin was not fully explored in our study and future research should focus on this aspect with more fine-grained analysis. Applying a relational event model (e.g., [38–40]) to Bitcoin, for instance, would improve our understanding of the endogenous and exogenous forces beyond its evolution. Furthermore, while our longitudinal setting based on annual network snapshots offers a good balance between length and granularity, it may hinder the identification of some short-term trends and patterns. Using shorter time intervals would also help to better account for fluctuations in network dynamics at a finer resolution. This would allow to focus on specific periods of extremely high price volatility, for example, to study network dynamics in such contexts, which may show unusual behaviour. This is because a longer observation period, as we considered in our study, may capture long-term trends, but miss the specificities of different events that could affect the Bitcoin network.

In our analysis, we have treated Bitcoin addresses as micro-level units of observation, without applying address clustering heuristics (e.g., [8]) or focusing on address reuse. While this approach has certain limitations -since multiple addresses may belong to the same entity- previous studies have shown that the results tend to remain consistent, regardless of whether clustering is applied (see, for example, [11] and [7]). Furthermore, the reliability of clustering heuristics is called into question by the presence of mixing services. Nevertheless, future research could benefit from systematically comparing both approaches, leveraging the growing availability of data and improved methodologies.

Finally, cryptocurrencies have become important assets, with expectations surrounding their potential role in the design of new market infrastructures for public purposes (e.g. [41, 42]). Therefore, there is a need for a more context-specific regulatory framework to increase the long-term sustainability of these new infrastructures [43]. Our findings on the endogenous centralisation of power through the concentration of wealth and activity show that the system is increasingly dominated by a few actors who could exploit their position at the expense of other smaller users. For instance, entities controlling a substantial share of tokens could impose barriers to entry, thereby undermining accessibility and long-term sustainability. As that the structure and dynamics of these cryptocurrencies are similar to those of traditional financial markets [44], policymakers and authorities could perhaps consider implementing policies and regulations that have been effective in increasing transparency and reducing concentration and inequality in financial markets. Furthermore, the position of cryptocurrencies within the global financial infrastructure means that the high concentration of resources and the network interconnectedness represent potential bottlenecks and channels for systemic risk propagation. Addressing these potential vulnerabilities is essential for ensuring the resilience, fairness, and long-term sustainability of cryptocurrency-based financial infrastructures.

## Acknowledgements

We would like to thank Carmen Guarner for her preliminary work on data extraction and pre-processing and for her help with the first steps of our study.

## Author contributions

**Conceptualization:** Marco Venturini, Flaminio Squazzoni.

**Data curation:** Marco Venturini, Daniel García-Costa, Elena Alvarez-García, Francisco Grimaldo.

**Formal analysis:** Marco Venturini, Daniel García-Costa, Elena Alvarez-García, Francisco Grimaldo, Flaminio Squazzoni.

**Methodology:** Marco Venturini, Flaminio Squazzoni.

**Visualization:** Marco Venturini.

**Writing – original draft:** Marco Venturini, Flaminio Squazzoni.

**Writing – review & editing:** Marco Venturini, Daniel García-Costa, Elena Alvarez-García, Francisco Grimaldo, Flaminio Squazzoni.

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
