## [Decision Letter · Decision Letter 0]

24 Jun 2025

PONE-D-25-21867Mapping network structures and dynamics of decentralised cryptocurrencies: The evolution of Bitcoin (2009-2023)PLOS ONE

Dear Dr. Venturini,

Thank you for submitting your manuscript to PLOS ONE. After careful consideration, we feel that it has merit but does not fully meet PLOS ONE’s publication criteria as it currently stands. Therefore, we invite you to submit a revised version of the manuscript that addresses the points raised during the review process.

**As per the review comments provided by the reviewer, you are advised to incorporate changes and submit the revised manuscript for consideration.**==============================

We look forward to receiving your revised manuscript.

Kind regards,

Nishi Malhotra

Academic Editor

PLOS ONE

Journal Requirements: 

Reviewers' comments:

Reviewer's Responses to Questions

**Comments to the Author**

1. Is the manuscript technically sound, and do the data support the conclusions?

Reviewer #1: Yes

Reviewer #2: Yes

2. Has the statistical analysis been performed appropriately and rigorously? 

Reviewer #1: Yes

Reviewer #2: Yes

3. Have the authors made all data underlying the findings in their manuscript fully available?

Reviewer #1: Yes

Reviewer #2: Yes

4. Is the manuscript presented in an intelligible fashion and written in standard English?

Reviewer #1: Yes

Reviewer #2: Yes

5. Review Comments to the Author

Reviewer #1: The manuscript "Mapping network structures and dynamics of decentralized cryptocurrencies: the evolution of Bitcoin (2009-2023)" is a well-crafted, comprehensive and thorough study that analyses the evolution of the Bitcoin transaction network from a network science perspective over fifteen years. The authors provide an exceptionally detailed analysis of the evolution of centralization, wealth concentration, and network structure, methodologically building on best practices while extending previous research.

Strengths:

The use of a complete blockchain dataset and its transparent processability (including available data for replication).

The longitudinal approach (annual snapshots) allows for tracking network evolution and trends over time.

Clear and consistent support for the rich-get-richer hypothesis and identification of three phases of development (Exploration, Adaptation, and Maturity).

The work is replicable, well documented and based on standard and credible metrics (degree distribution, Gini, assortativity, SCC/WCC, transitivity, etc.).

Minor modifications recommended:

Discussion of implications: The results are relevant to current debates on decentralization and regulation of cryptocurrencies. The discussion could be strengthened by a more critical reflection on these implications, especially in light of the endogenous concentration of power found.

Limitations section: The limitations of the paper (e.g., use of annual cross-sections, omission of subtle temporal fluctuations) merit their own clearer section.

Readability to a broader audience: PLOS ONE has a wide readership. Some terms, such as "transitivity index" and "weakly vs. strongly connected components", would benefit from a brief explanation.

Visualizations: Several figures (especially Figs. 1-2 and Figs. 5-6) deserve to be accompanied by more prominent captions or legends to be interpreted independently outside the main text.

Emphasis on originality: Although the paper refers to relevant literature (e.g., Kondor, Maesa), I recommend that it be made even more explicit what makes this paper different and pushes the boundaries of previous research.

Overall, this very high-quality study deserves to be published after minor additions. The authors have done an excellent job handling a huge dataset and presenting important conclusions for understanding the evolution of decentralized systems.

Reviewer #2: General Assessment:

This manuscript presents a comprehensive longitudinal analysis of the Bitcoin transaction network from 2009 to 2023. It is a valuable and technically sound contribution that extends previous work by nearly a decade and offers new insights into the structural evolution of the Bitcoin ecosystem. The division of the network's history into three evolutionary phases : Exploration, Adaptation, and Maturity, is well argued and supported by both empirical metrics and historical context.

Technical Soundness:

The study’s methodology is rigorous and replicable. The authors use a large, reliable dataset, implement appropriate filters (e.g., dust thresholds), and exclude irrelevant system artifacts. The definition of network graphs and use of weighted edges is clearly explained and justified. The segmentation of network snapshots into annual intervals helps capture dynamics without overgeneralizing.

Statistical and Network Analysis:

The network metrics (degree distributions, Gini coefficients, assortativity, clustering, and component analyses) are all appropriate and well supported. The rich-get-richer hypothesis is tested with clear methodology, consistent with prior work, and expanded to include updated data post-2015. The analysis of node persistence and inequality provides strong evidence of path dependency and centralisation.

Writing and Clarity:

The manuscript is well-written, logically organized, and accessible to both technical and interdisciplinary audiences. While the introduction includes some speculative references to political events (e.g., US presidential election), these do not undermine the scientific integrity of the analysis. Still, the authors may consider rephrasing these in a more neutral tone.

Data and Reproducibility:

The data availability is excellent. Public blockchain data is properly referenced, and the derived dataset is deposited with a DOI, fully meeting PLOS ONE's open data requirements.

Suggestions for Improvement:

Add a brief discussion of limitations regarding address-level analysis (e.g., address reuse, clustering heuristics).

Consider a diagram or figure to visualize the data pipeline from raw blockchain data to network construction.

Tidy up punctuation around compound terms and ensure consistent hyphenation and spacing (e.g., “Exploration, Adaptation and Maturity” - “Exploration, Adaptation, and Maturity”).

Clarify briefly how some of the early richest nodes were de-anonymized despite limited historic metadata.

Conclusion:

This is a strong, well-executed manuscript that provides valuable insights into Bitcoin’s structural evolution. With minor revisions, it will make a meaningful contribution to the literature on cryptocurrency networks and decentralisation.

6. PLOS authors have the option to publish the peer review history of their article (what does this mean?). If published, this will include your full peer review and any attached files.

Reviewer #1: No

Reviewer #2: **Yes: **Shreya Macherla

---

## [Author Response · Author response to Decision Letter 1]

7 Jul 2025

Dear PLoS ONE editor,

Thank you for giving us the opportunity to revise and resubmit our manuscript. We have carefully considered all reviewer comments and added point-to-point responses to their requests in the letter of response.

For the points mentioned in your letter, we respond here.

- Thank you for uploading your study's underlying data set. Unfortunately, the repository you have noted in your Data Availability statement does not qualify as an acceptable data repository according to PLOS's standards. At this time, please upload the minimal data set necessary to replicate your study's findings to a stable, public repository (such as figshare or Dryad) and provide us with the relevant URLs, DOIs, or accession numbers that may be used to access these data. For a list of recommended repositories and additional information on PLOS standards for data deposition, please see https://journals.plos.org/plosone/s/recommended-repositories.

Author: After a second check, it is clear that the Unimi Dataverse fully complies with PLOS’s standards as it adheres to Fairsharing principles, it is included in the Registery of Research Data Repositories (Re3Data), and follows adequate CC By licenses.

Author: No retracted articles are included.

Thank you again for handling our manuscript ensuring timely and informative reports.

We sincerely hope that now the manuscript is acceptable for publication in your journal.

Rebuttal letter, response to the editor and reviewers

Reviewer #1

Minor modifications recommended:

- Discussion of implications: The results are relevant to current debates on decentralization and regulation of cryptocurrencies. The discussion could be strengthened by a more critical reflection on these implications, especially in light of the endogenous concentration of power found.

Author: Thank you for this suggestion. We have added a more comprehensive reflection on the implications of the endogenous concentration of wealth and power.

- Limitations section: The limitations of the paper (e.g., use of annual cross-sections, omission of subtle temporal fluctuations) merit their own clearer section.

Author: Thank you for this suggestion. We have now elaborated on the study limitations and explored future developments.

- Readability to a broader audience: PLOS ONE has a wide readership. Some terms, such as "transitivity index" and "weakly vs. strongly connected components", would benefit from a brief explanation.

Author: Good point. We have accompanied technical terms with plain language explanations.

- Visualizations: Several figures (especially Figs. 1-2 and Figs. 5-6) deserve to be accompanied by more prominent captions or legends to be interpreted independently outside the main text.

Author: Good point. We have extended and integrated the captions, as suggested.

- Emphasis on originality: Although the paper refers to relevant literature (e.g., Kondor, Maesa), I recommend that it be made even more explicit what makes this paper different and pushes the boundaries of previous research.

Author: Thank you for this remark. We have expanded on the relevance of our study in the initial section. We highlighted that our contribution consists of three main points: first, we have updated previous studies, which were limited to a few years of observation; second, we have offered a new categorisation of the Bitcoin’s life phases by capturing internal dynamics and external evolution (both points have been ignored in previous research); third, we provided a broad range of measures to disentangle the complexity of the system, while previous research tended to focus on specific aspects only (e.g., power law degree distributions). Furthermore, we tested for the rich-get-richer mechanism to include post-2015 data and we focused on centralisation, node persistence and wealth inequality. These points have been better specified in the revised version of the text.

Reviewer #2

Minor modifications recommended:

- The manuscript is well-written, logically organized, and accessible to both technical and interdisciplinary audiences. While the introduction includes some speculative references to political events (e.g., US presidential election), these do not undermine the scientific integrity of the analysis. Still, the authors may consider rephrasing these in a more neutral tone.

Author: Good point. However, there was no political intention behind as we simply tried to give context and ground our topic in the actual political scenario. But you are right, in these difficult times, better not to convey any political message and do an extra-effort to be as neutral as possible.

- Add a brief discussion of limitations regarding address-level analysis (e.g., address reuse, clustering heuristics).

Author: Good point. Following also a similar request by Reviewer 1, we have expanded on the study limits and included also this point.

- Consider a diagram or figure to visualize the data pipeline from raw blockchain data to network construction.

Author: Good point. We have added a diagram to visualise the data pipeline.

- Tidy up punctuation around compound terms and ensure consistent hyphenation and spacing (e.g., “Exploration, Adaptation and Maturity” - “Exploration, Adaptation, and Maturity”).

- Clarify briefly how some of the early richest nodes were de-anonymized despite limited historic metadata.

Author: Thank you for your suggestions. We have added clarifications to explain the point. Most of the deanonymisation techniques consisted of web scraping and searching for digital traces (e.g., some users publicly state their address in forums). Nonetheless, we acknowledge that only a small portion of addresses has been de-anonymised, almost only exchanges and big players. As clarified in the revised version of the text, we relied on previous works and Arkham Intelligence and BitInfoCharts.

Best regards

---

## [Decision Letter · Decision Letter 1]

21 Jul 2025

Mapping network structures and dynamics of decentralised cryptocurrencies: The evolution of Bitcoin (2009-2023)

PONE-D-25-21867R1

Dear Dr Nishi Malhotra 

We’re pleased to inform you that your manuscript has been judged scientifically suitable for publication and will be formally accepted for publication once it meets all outstanding technical requirements.

Kind regards,

Nishi Malhotra

Academic Editor

PLOS ONE

Additional Editor Comments (optional):

Reviewers' comments:

Reviewer's Responses to Questions

**Comments to the Author**

1. If the authors have adequately addressed your comments raised in a previous round of review and you feel that this manuscript is now acceptable for publication, you may indicate that here to bypass the “Comments to the Author” section, enter your conflict of interest statement in the “Confidential to Editor” section, and submit your "Accept" recommendation.

Reviewer #1: All comments have been addressed

Reviewer #2: All comments have been addressed

2. Is the manuscript technically sound, and do the data support the conclusions?

Reviewer #1: Yes

Reviewer #2: (No Response)

3. Has the statistical analysis been performed appropriately and rigorously? 

Reviewer #1: Yes

Reviewer #2: (No Response)

4. Have the authors made all data underlying the findings in their manuscript fully available?

Reviewer #1: Yes

Reviewer #2: (No Response)

5. Is the manuscript presented in an intelligible fashion and written in standard English?

Reviewer #1: Yes

Reviewer #2: (No Response)

6. Review Comments to the Author

Reviewer #1: I want to thank the authors for their high-quality and thorough revision of the manuscript. All comments have been addressed substantively – the discussion on the consequences of centralisation has been expanded, the study's limitations have been identified, and technical terms have been made accessible to a broader audience. I also appreciate the revision of the image captions and the more precise emphasis on the article's originality in relation to previous literature. I consider the revised manuscript ready for publication.

Reviewer #2: (No Response)

7. PLOS authors have the option to publish the peer review history of their article (what does this mean?). If published, this will include your full peer review and any attached files.

Reviewer #1: No

Reviewer #2: **Yes: **Shreya Macherla

---

## [Editor Report · Acceptance letter]

PONE-D-25-21867R1

PLOS ONE

Dear Dr. Venturini,

I'm pleased to inform you that your manuscript has been deemed suitable for publication in PLOS ONE. Congratulations! Your manuscript is now being handed over to our production team.

Kind regards,

on behalf of

Dr. Nishi Malhotra

Academic Editor

PLOS ONE